

# Intrinsic RGB and multispectral images recovery by independent quadratic programming

Alexandre Krebs, Yannick Benezeth and Franck Marzani

ImViA EA7535, Université de Bourgogne Franche-Comté, Dijon, France

## ABSTRACT

This work introduces a method to estimate reflectance, shading, and specularity from a single image. Reflectance, shading, and specularity are intrinsic images derived from the dichromatic model. Estimation of these intrinsic images has many applications in computer vision such as shape recovery, specularity removal, segmentation, or classification. The proposed method allows for recovering the dichromatic model parameters thanks to two independent quadratic programming steps. Compared to the state of the art in this domain, our approach has the advantage to address a complex inverse problem into two parallelizable optimization steps that are easy to solve and do not require learning. The proposed method is an extension of a previous algorithm that is rewritten to be numerically more stable, has better quantitative and qualitative results, and applies to multispectral images. The proposed method is assessed qualitatively and quantitatively on standard RGB and multispectral datasets.

## INTRODUCTION

Light reflected on the surface of an object could be either diffuse or specular. Diffuse reflection is produced by rough surfaces that tend to reflect light in all directions while specular reflection is defined as light reflected at a definite angle, like a mirror reflection. These two phenomena appear on an image, thus, a challenging task is to isolate their contributions. Shape, color, and geometry are very useful information that could be obtained from the decomposition of diffuse and specular reflection. For example, the color of an object can be used for segmentation, classification, or recoloring and the shape and the geometry gives 3D information about the environment and could be used for object recognition. Several models have been proposed to model light reflected on a surface. One of the simplest models is the Lambertian model proposed by Lambert in 1760. The model is expressed by

$$I(u, \lambda) = \frac{1}{\pi} l(\lambda) S(u, \lambda) \cos(\theta_i) dw_i \qquad (1)$$

where $I(u, \lambda)$ is the diffuse radiance at pixel $u$ and wavelength $\lambda$, $S(u, \lambda)$ is the surface reflectance, $l$ is the light source radiance, $\theta_i$ is the incident angle and $dw_i$ is the solid angle of the light source viewed from pixel $u$.

Corresponding author
Alexandre Krebs,
alexandre.krebs@u-bourgogne.fr

The dichromatic model is also widely used in the literature. It was first proposed by *Shafer (1985)* for modeling dielectric objects. The model is mathematically defined by the equation:

$$I(u, \lambda) = l(\lambda)(g(u)S(u, \lambda) + k(u)) \tag{2}$$

where $I$, $l$, and $S$ are defined as previously, $g$ is the shading factor and $k$ is the specular coefficient. The dichromatic model assumes uniform illumination across the spatial domain as explained in the book of *Robles-Kelly & Huynh (2012)*. In this model, an image can be split into a diffuse and a specular part. The shading factor $g$ governs the proportion of diffuse light reflected from the object and $k$ models the irregularities of the micro-facet structure that causes specularity in the scene. Compared to the lambertian model, the dichromatic model adds the specular part $k$ and for a purely diffuse lambertian surface, $g(u) = \frac{1}{\pi}\cos(\theta_i)dw_i$ and $I(u, \lambda) = l(\lambda)g(u)S(u, \lambda)$.

Other models exist like Shape, illumination and reflectance from shading (SIRFS), developed by *Barron & Malik (2015)*. The model is parametrized with a *rendering engine* and a spherical harmonic model of illumination. To the contrary of the dichromatic model, SIRFS is based on computer graphics and not on phenomenology. All these models can be used to decompose an image or to generate synthetic images.

Methods to inverse the dichromatic model are often based on the neighborhood analysis of each pixel: *Tan, Nishino & Ikeuchi (2004)* and *Tan & Ikeuchi (2005)* have described a specular-to-diffuse mechanism that is applied to local neighborhoods having the same reflectance. Fast recovery of intrinsic images from a single image already exists. *Yoon, Choi & Kweon (2006)* create specular-free 2-channel images and *Yang, Tang & Ahuja (2015)* use guided filtering (originally proposed by *He, Sun & Tang (2013)*) to remove specularity. Recent progress in deep learning encourages researchers in the field to use convolutional neural network (CNN) based approaches to solve the inversion problem like *Son & Lee (2016)* or *Shi et al. (2017)*. Few works consider a non-local strategy like *Xie et al. (2016)*. They encourage distant clusters that have the same color to have the same reflectance. *Shen, Tan & Lin (2008)* use intensity-normalized color information as texture vectors and encourage distant pixels that have the same texture vectors to have the same reflectance.

The decomposition of multispectral images into photometric invariants is recent. For example, *Huynh & Robles-Kelly (2008, 2010)* have worked on multispectral images. Their method consists of minimizing objective functions based on the dichromatic model to recover intrinsic images. The decomposition was then used for skin recognition, material clustering, and specularity removal. *Koirala et al. (2011)* have another approach. They detect and remove specularity with a filter that coefficients are found by constrained energy minimization. The dichromatic model parameters recovery can also be achieved with the inversion of a linear model as demonstrated by *Fu, Tan & Caelli (2006)*. They have applied Orthogonal Subspace Projection to remove specularity. Similarly, *Zheng, Sato & Sato (2015)* and *Chen, Drew & Li (2017)* separate the illumination

spectra from the reflectance using low-rank matrix factorization following the Lambertian model.

The purpose of this work is to inverse the dichromatic model. Precisely, three photometric invariants $g$, $S$, and $k$ are recovered. These photometric invariants are recovered thanks to two quadratic programming steps. The presented inversion method has the advantages to be learning free and applies to RGB images as well as multispectral images. Thus, the rest of the paper is organized as follows: in "Underdetermination and Related Work," the inversion problem and its underdetermination are analyzed. The solutions and the limitations found in the literature are explored to compare our approach to existing methods. Then, the proposed method is detailed in "Method." Finally, we assess the robustness of the proposed method qualitatively and quantitatively in "Results and Discussion."

## UNDERDETERMINATION AND RELATED WORK

Even if the dichromatic model is rather simple, its inversion is still complex. The inversion process is an underdetermined problem. One single image could have been obtained by a large combination of illumination, shape, and reflectance.

Mathematically, we note that in Eq. (2), there could be any balancing factor between $g$ and $S$ that is, if $S^*$ and $g^*$ are solutions to the inverse problem, then $\alpha S^*$ and $\frac{g^*}{\alpha}$ are also solutions to the inverse problem for any positive scalar $\alpha$. From a numerical point of view, there are fewer equations than unknowns. Let us define $N_p$ as the number of pixels and $N_c$ as the number of wavelengths. According to Eq. (2), there are $N_p \times N_c$ equations for $N_c + N_p + N_p \times N_c + N_p$ unknowns. This comparison shows clearly that the problem is underdetermined, thus the inversion algorithm should include soft or hard constraints to overcome the underdetermination. For example, $g$, $S$, and $k$ have a physical meaning, they must be positive numbers.

One of the simplest ways to reduce the number of unknowns is to assume that the illumination spectrum is known or can be experimentally estimated. This can be done by imaging a white standard reference and define $l$ as the spatial mean spectra. $l$ can also be obtained thanks to one of the reference methods taken from literature like the White-Patch method, the Grey-World method, or the Grey-Edge method as explained by *Huynh & Robles-Kelly (2010)*. Recent deep-learning-based algorithms compete with these methods like the CNN of *Bianco, Cusano & Schettini (2015)* or the mixed pooling neural networks of *Fourure et al. (2016)*. Once the spectrum of the illumination is known, the Eq. (2) becomes:

$$R(u, \lambda) = \frac{I(u, \lambda)}{l(\lambda)} = g(u)S(u, \lambda) + k(u). \tag{3}$$

This simplification suggests that the reflectance spectra $S$ is related to $R$ with a scalar and an offset.

For some applications, if only one material is considered, the number of unknowns can be further reduced because the reflectance $S$ is no more pixel dependent and the Eq. (3) is simplified by

$$R(u, \lambda) = g(u)S(\lambda) + k(u). \tag{4}$$

This system is overdetermined and can be solved by linear regression as suggested by *Robles-Kelly & Huynh (2012)*. Unfortunately, the uniqueness of the material is a strong assumption and is rarely applicable in practice.

*Barron & Malik (2012, 2015)* have expressed priors on the illumination, the reflectance and the shape of an object. These constraints are soft constraints and even if we are not using the same model, we use some of their ideas to build our optimization algorithm. For example, Barron et al. explain that surfaces tend to be smooth, thus the shading image *g* is also smooth. This assumption is also used in the papers of *Gu & Robles-Kelly (2016)* and *Huynh & Robles-Kelly (2008)*. The smoothness is expressed by a regularization term that minimizes the gradient and the mean curvature respectively. *Barron & Malik (2015)* also expressed the fact that the number of different reflectances in an image tends to be small. This means that the palette used for an image is small. These priors reflect good assumptions but require learning. In this case, learning-based methods would require a lot of training samples and would be dependent on the number of channels of the image. In practice *Barron & Malik (2015)* have trained their priors for gray-scale image (one channel) and RGB images (three channels) independently.

Some works overcome the underdetermination by increasing the amount of data available, for example by combining multiple views of the scene. Using multiple images makes easier the separation of diffuse and specular components. For example, *Umeyama & Godin (2004)* use a rotating polarizer to acquire several images and then apply Independent component analysis assuming the probabilistic independence between diffuse and specular components. *Feris et al. (2004)* use multi-flash images to reduce specularity and *Xie et al. (2016)* use stereoscopic images and inverse the Lambertian model $R(u, \lambda) = g(u)S(u, \lambda)$. *Zhou, Krähenbühl & Efros (2015)* increases the amount of information by asking the user to order image patches according to their brightness, thus producing a data-driven reflectance prior. The use of several views of the same scene decreases the underdetermination but is also more cumbersome.

The proposed method takes into account the numerical constraints that were observed. As most of the real-life objects are smooth, a soft smoothness is also introduced. The illumination spectrum is known thus, the efforts are focused on the resolution of Eq. (3). The next section details the complete method to recover the intrinsic images and explains the constraints that were used.

## METHOD

In a previous paper, we have proposed a learning free method to solve the decomposition problem with two quadratic programming steps (*Krebs, Benezeth & Marzani (2017)*). The shading factor *g* and the specular image *k* were indirectly recovered as the minimum of

quadratic objective functions subject to linear constraints. It means that the decomposition was obtained by solving two problems under the general form:

$$\begin{cases} \mathbf{x}^* = \underset{\mathbf{x}}{\text{argmin}} \ \dfrac{1}{2}\mathbf{x}^t\mathbf{Q}\mathbf{x} + \mathbf{c}^t\mathbf{x} \\ \text{subject to} \\ \mathbf{A}\mathbf{x}^* \leq \mathbf{b} \\ \mathbf{A_{eq}}\mathbf{x}^* = \mathbf{b_{eq}} \end{cases} \tag{5}$$

where $\mathbf{x}^*$ is the desired solution vector (i.e., the flattened version of $g$ or indirectly $k$). $\mathbf{Q}$, $\mathbf{A}$, and $\mathbf{A_{eq}}$ are matrices. The values of $\mathbf{Q}$ take into account the similarity between neighboring pixels of the input image. $\mathbf{c}$, $\mathbf{b}$, and $\mathbf{b_{eq}}$ are column vectors, with the same number of elements as the number of pixels of the input image.

Quadratic programming is the process of solving this kind of optimization problem. Nowadays, these problems are well known and it exists a variety of methods to solve them like the interior point, the active set, the augmented Lagrangian, or the conjugate gradient detailed by *Nocedal & Wright (2006)*.

The previous method was based on the distinction of three cases, the case where neighboring pixels belong to the same material, the case where they are not and the case where the pixels are gray.

In the following parts, an improved version of our algorithm written in 2017 is introduced. The goal is to get rid of some drawbacks while staying learning free and preserving the quadratic formulation to keep the simplicity of resolution. The objective functions are changed and expressed in the logarithmic domain. Working in the logarithmic domain allows us to reach better numerical stability because it transforms divisions into subtractions. This leads to less instability when dividing by numbers close to zero and thus, much better quantitative and qualitative results. Moreover, the algorithm is extended to multispectral images. To the best of our knowledge, no other method that considers RGB and multispectral images have been published.

**Indirect recovery of the shading factor**

The first goal of the proposed method is to recover the shading factor $g$. $g$ is actually indirectly recovered: we define the unknown of the optimization problem as $x = \ln(g)$.

It has been shown that if two pixels $u$ and $v$ belong to the same material, the ratio between $g_u$ and $g_v$ (values of the image $g$ at pixels $u$ and $v$) is equal to the ratio between $\sigma_u$ and $\sigma_v$ (the standard deviations of $R$ along the wavelengths axis):

$$\frac{g_u}{g_v} = \frac{\sigma_u}{\sigma_v} \tag{6}$$

because

$$\frac{\sigma_u}{\sigma_v} = \frac{std(R_u, \lambda)}{std(R_v, \lambda)} = \frac{g_u std(S_u, \lambda)}{g_v std(S_v, \lambda)} = \frac{g_u}{g_v}. \tag{7}$$

Thus, applying the logarithm transforms ratios into differences

$$x_u - x_v = \ln(\sigma_u) - \ln(\sigma_v). \tag{8}$$

The key idea is then to write an objective function as a weighted sum of squared residuals:

$$f_1(x) = \sum_u \sum_{v \in \mathcal{N}(u)} \zeta_{u,v} (x_u - x_v - \ln(\sigma_u) + \ln(\sigma_v))^2 \tag{9}$$

where $\mathcal{N}(u)$ denotes the neighborhood of $u$, $\zeta_{u,v}$ is a weight between 0 and 1 corresponding to the similarity measure between two spectra at pixels $u$ and $v$:

$$\zeta_{u,v} = \exp\left(\frac{-SAM(R_u, R_v)^2}{r}\right) \tag{10}$$

with $r$ as the bandwidth parameter and SAM as the Spectral angle mapper, one spectral similarity measure explored by *Galal, Hasan & Imam (2012)*.

$$SAM(R_u, R_v) = \arccos\left(\frac{\sum_{i=1}^{N_c} R_{ui} R_{vi}}{\sqrt{\sum_{i=1}^{N_c} R_{ui}^2} \sqrt{\sum_{i=1}^{N_c} R_{vi}^2}}\right). \tag{11}$$

In case pixels $u$ and $v$ do not belong to the same material (i.e., $\zeta_{u,v}$ is close to zero), $x$ is assumed to be smooth i.e., $x_u \approx x_v$. The complementary objective function is thus created under the form:

$$f_2(x) = \sum_u \sum_{v \in \mathcal{N}(u)} (1 - \zeta_{u,v})(x_u - x_v)^2. \tag{12}$$

From an image processing point of view, minimizing the distance between $x_u$ and $x_v$ is intuitively like applying an averaging filter on $x$ which is also equivalent to applying a geometric mean filter on $g$. This smoothing is more robust to positive outliers than the classical averaging filter.

There is a third case which is more difficult. If two pixels $u$ and $v$ are gray, then the standard deviation is close or equal to zero and thus $f_1$ could be unstable. A gray pixel is defined as a pixel for which $R_u$ is nearly constant for every wavelength and thus cannot be separated into a diffuse part and a specular part. In this case, the mean over the wavelengths $\mu$ is used instead of the standard deviation $\sigma$.

This case corresponds to the assumption that there is no specularity on gray objects (i.e., that $k = 0$). A third part of the objective function is thus written:

$$f_3(x) = \sum_u \sum_{v \in \mathcal{N}(u)} \zeta_{u,v} \zeta_u \zeta_v (x_u - x_v - \ln(\mu_u) + \ln(\mu_v))^2 \tag{13}$$

New symbols are introduced with the following definition: $\zeta_u = \zeta(R_u, \mathbf{1})$ and $\overline{\zeta_u} = 1 - \zeta(R_v, \mathbf{1})$ ($\mathbf{1}$ being a vector of ones). $\zeta_u$ is an indicator, based on SAM metric, that emphasizes the spectra that are nearly gray (value close to 1) or not (value close to 0).

On the same time, $f_1$ and $f_2$ are slightly modified with an additional factor that discards gray pixels:

$$f_1(x) = \sum_u \sum_{v \in \mathcal{N}(u)} \zeta_{u,v} \overline{\zeta_u \zeta_v} (x_u - x_v - \ln(\sigma_u) + \ln(\sigma_v))^2, \tag{14}$$

$$f_2(x) = \sum_u \sum_{v \in \mathcal{N}(u)} (1 - \zeta_{u,v} \zeta_u \zeta_v - \zeta_{u,v} \overline{\zeta_u \zeta_v})(x_u - x_v)^2. \tag{15}$$

The final objective function is written as the sum of $f_1, f_2$, and $f_3$ so that all three cases are well encompassed.

$$f_{shading}(x) = f_1(x) + f_2(x) + f_3(x). \tag{16}$$

As this objective function is the sum of quadratic functions, $f_{shading}$ is also quadratic and its minimization can be seen as a quadratic programming task as presented in the system (5).

Once the objective function is built, hard constraints are considered. Mathematically, we have seen in "Underdetermination and Related Work" that there could be any balancing factor between $g$ and $S$. Thus, in the logarithmic domain, considering $x^*$ instead of $g^*$, there can be any offset $\varepsilon$ we can add to $x^*$, the solution will still hold. Thus, we can constraint the sum of all elements of $x$ to be equal to an arbitrarily chosen constant $c$.

## Indirect recovery of the specular factor

The second goal of the proposed method is to recover the specular factor $k$. This part of the method is independent of the previous calculus of $g$, and thus, the two optimizations are perfectly interchangeable or can be parallelized to speed up the algorithm. The intermediate variable $y = \ln(g\overline{S})$ is computed as the minimum of a constrained quadratic objective function. The symbol $\overline{S}$ being the mean of $S$ over the wavelengths.

If two pixels $u$ and $v$ belong to the same material, then:

$$y_u - y_v = \ln(\sigma_u) - \ln(\sigma_v) \tag{17}$$

for the same reason as for Eq. (8).

Thus our objective function can be written as the square of the difference between $y_u$ and $y_v$ weighted by $\zeta_{u,v}$ to express the similarity of the material and $\overline{\zeta_u \zeta_v}$ to express the fact that the pixels cannot be gray:

$$f_4(y) = \sum_u \sum_{v \in \mathcal{N}(u)} \zeta_{u,v} \overline{\zeta_u \zeta_v} (y_u - y_v - \ln(\sigma_u) + \ln(\sigma_v))^2. \tag{18}$$

$f_4$ is analogous to the function $f_1$ in Eq. (14).

Like in previous part, a complementary function is written. In this case, we assume that the specularity is negligible compared to the diffuse part i.e., $y_u \approx \mu_u$.

$$f_5(y) = \sum_u \sum_{v \in \mathcal{N}(u)} (1 - \zeta_{u,v} \overline{\zeta_u \zeta_v})(y_u - y_v - \ln(\mu_u) + \ln(\mu_v))^2. \tag{19}$$

$f_4$ and $f_5$ are also quadratic and the final objective function can be written as:

$$f_{\text{specular}}(y) = f_4(y) + f_5(y). \tag{20}$$

Because of the physical constraint, $k$ is bounded below by 0, all elements of $k$ must be positive. Moreover, the consideration of the minimum of $R$ over the wavelength:

$$\min(R_u, \lambda) = g_u \min(S_u, \lambda) + k_u \tag{21}$$

shows that $k_u$ is also upper-bounded by $\min(R_u, \lambda)$. Thus, $y_u$ is also bounded:

$$\ln(\mu - \min(R_u, \lambda)) \leq y_u \leq \ln(\mu). \tag{22}$$

As for $x$, $y$ is recovered thanks to a quadratic programming algorithm. $g$, $k$, and $S$ are then obtained with

$g = \exp(x)$

$k = \mu - \exp(y)$

$$S = \frac{R - k}{g}. \tag{23}$$

To conclude on this section, two quadratic objective functions have been built allowing us to recover indirectly $g$, $k$, and $S$. Only simple statistical tools (standard deviation and mean) have been used making the method applicable to RGB images as well as multispectral images. These functions decompose all pixels into three categories: neighboring pixels that belong to the same material, pixels from different materials and gray pixels.

## RESULTS AND DISCUSSION

This section presents the qualitative and quantitative results of the presented method. First, qualitative results are presented and then, metrics are introduced to assess the quality of the method quantitatively. It is very important to compare our work to current methods in the literature, thus, in the following parts, the references *Barron & Malik (2015)*, *Yang, Wang & Ahuja (2010)*, *Gu, Robles-Kelly & Zhou (2013)*, *Huynh & Robles-Kelly (2010)*, and the previous version of the algorithm (*Krebs, Benezeth & Marzani (2017)*) are used as comparative methods. Afterward, we will use the following abbreviations to refer to each of these methods:

- *LS* for *Gu, Robles-Kelly & Zhou (2013)*
- *KL* for *Huynh & Robles-Kelly (2010)*
- *Barron* for *Barron & Malik (2015)*
- *Yang* for *Yang, Wang & Ahuja (2010)*
- *Krebs* for *Krebs, Benezeth & Marzani (2017)*

*LS* employs shapelets to recover the shading of an image. *KL* is based on objective functions with a regularization term that enforces the smoothness of $g$. *Barron* uses priors

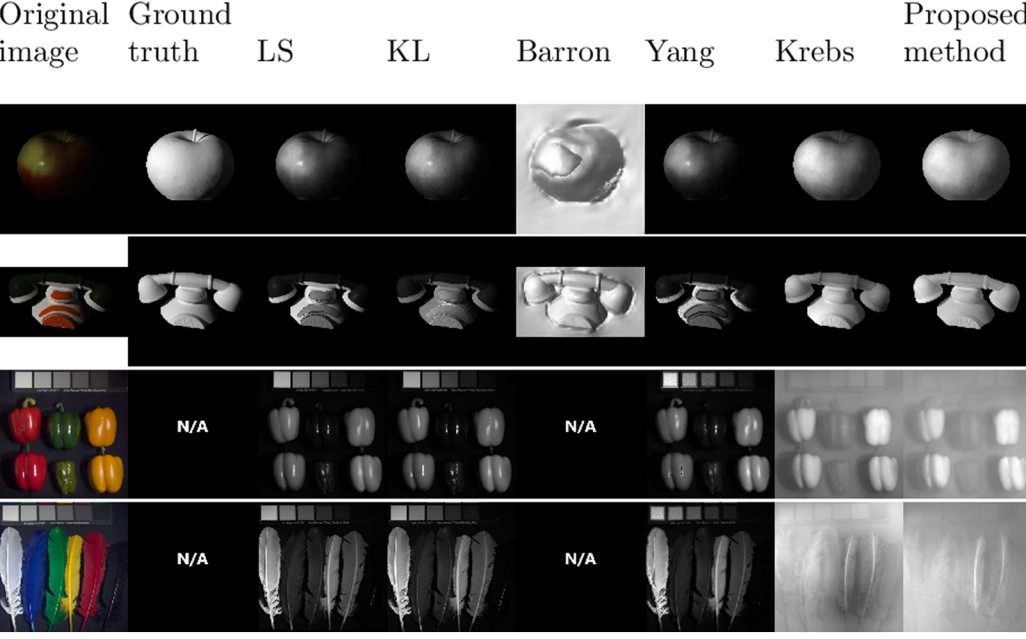

**Figure 1 Results of shading images for the different methods: from left to right, the original image, the ground truth, _LS, KL, Barron, Yang, Krebs_, and the proposed method.**

to recover the most probable illumination, shape, and reflectance, and _Yang_ uses guided image filtering to iteratively remove specularity.

We have tested the different algorithms on the Massachusetts Institute of Technology (MIT) intrinsic dataset created by _Grosse et al. (2009)_. This dataset provides 20 images along with a ground truth image for the reflectance $S$, the shading $g$, and the specularity $k$, namely $S_{\text{true}}$, $g_{\text{true}}$, and $k_{\text{true}}$. We have also applied our algorithm on the CAVE (Computer Vision Laboratory at Columbia University) Multispectral image dataset which provides multispectral images without ground truth. The dataset is available thanks to _Yasuma et al. (2010)_.

For visualization purposes, the multispectral images are transformed to RGB images via a multilinear transformation. All images are padded with black pixels to be square and scaled. All images are also divided by the illumination spectrum. For the MIT database, we assume $l$ is white and for the CAVE database, $l$ is obtained thanks to the white patch of ColorChecker appearing on each image.

## Qualitative results

Figure 1 presents shading images $g$ resulting from all methods on four examples. The two first rows are images coming from the MIT dataset, the two other rows are multispectral images from the CAVE dataset. The first column contains the input image, the second one contains the ground truth shading images. The next columns are respectively the results given by _LS, KL, Barron, Yang, Krebs_ and the proposed method. Considering the apple, the specular spot still appears for _LS, KL_, and _Yang_ while the proposed method is more robust and is not corrupted by specularity. For the phone, strong gradients appear

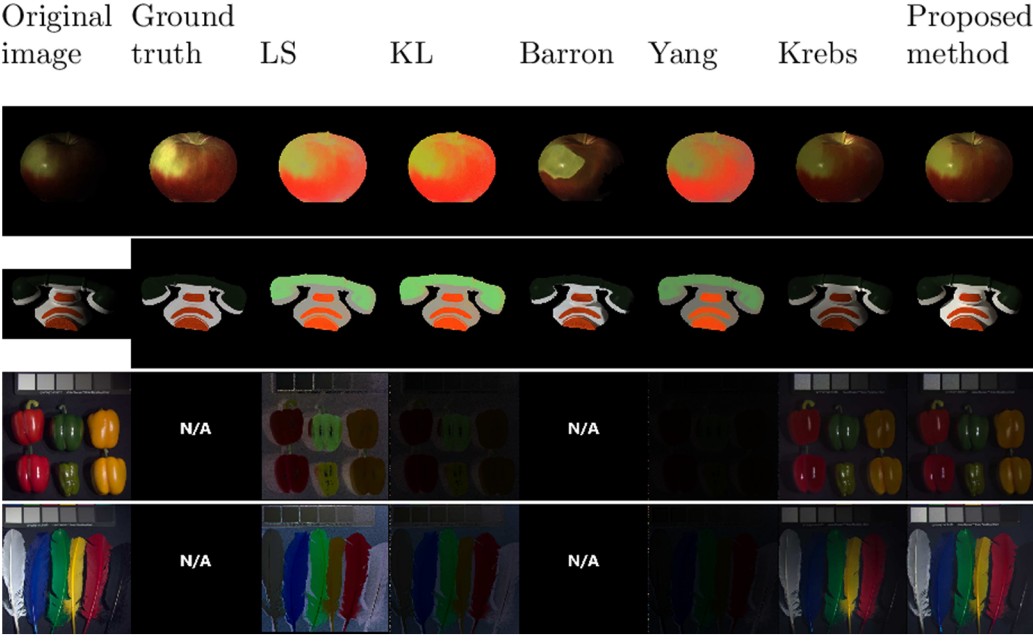

**Figure 2 Results of reflectance images for the different methods: from left to right, the original image, the ground truth, *LS, KL, Barron, Yang, Krebs*, and the proposed method.**

for methods *LS*, *KL*, and *Yang* that are not due to the shape of the object but are induced by color changes and thus should not appear. The smoothness term introduced in Eq. (15) makes the difference with other recent works. The key is that this term only acts on color gradients and not on uniform areas. On the multispectral peppers image, we can see that the specularity is successfully removed while it is still appearing with the other methods. The shading image *g* for the colored feathers seems flat and blurry. But, as this scene is flat, *g* is also flat. Still, the method tends to oversmooth the shading image on multispectral images. This is because the smoothness term in Eq. (15) is a good assumption if there is only one object on the image which is not the case on the images from the CAVE dataset (six peppers and six feathers for example).

In Fig. 2, results of reflectance images are also compared between all methods. The first column contains the input image, the second one contains the ground truth reflectance images. The next columns are the results obtained with *LS*, *KL*, *Barron*, *Yang*, *Krebs*, and the proposed method. The observation is that for the methods *LS*, *KL*, and *Yang*, the color are corrupted. These methods fail to conserve the good ratio between colors. For example, the white part of the phone looks gray for *LS*, *KL*, and *Yang*. That means that *g* was over-rated on the white part. Another manifestation of this non-conservation of color ratios can be seen with the feathers image: the first feather on the left should be white and appears black for all other methods. It is exactly the opposite with the first feather on the right which is black and appears white for the other methods. The smoothness constraint (15) helps also to keep the color ratio which is essential for the visualization of *S*. We can note that, on the case of white objects, the objective function (13) helps not to be confused with specularity. However, it can be seen that some specular is remaining in

**Table 1 Quantitative results on the MIT dataset.** The numbers written in bold represent the best scores among all compared methods.

|  | LS | KL | Barron | Yang | Krebs | Proposed method |
|---|---|---|---|---|---|---|
| Execution time (s) | 553 | 68 | 16348 | **9** | 22187 | 675 |
| SSE on normalized $g$ ($\times 10^{-5}$) | 0.780 | 0.504 | 1.765 | 0.883 | 0.384 | **0.183** |
| SSE on normalized $S$ ($\times 10^{-5}$) | 0.225 | 0.233 | 0.980 | 0.240 | 0.267 | **0.144** |
| MSE on $k$ ($\times 10^{8}$) | 0.102 | 1.223 | N/A | 0.206 | 0.053 | **0.007** |
| SAM on $S$ | 0.021 | 0.041 | 0.032 | 0.025 | 0.030 | **0.020** |

the reflectance images of the apple and the peppers. It appears in gray for *LS*, *KL*, and *Yang* and in white for *Barron*, *Krebs*, and the proposed method. This can be due to saturated pixels. The information of color under the specularity becomes too low to be recognized.

We do not present results on specular images $k$ because the images are very dark, the specularity is only the white spots we can see on objects (for example on the apple). Thus, excepting these white spots, the rest of the $k$ images are completely black and thus are difficult to compare.

## Quantitative results

Qualitative analysis is not sufficient to prove the robustness of the proposed method. Thus, metrics between $g$, $S$, and $k$ are computed.

As explained in part 2 there can be any scaling factor between $g$ and $S$. Thus, we need to normalize $g$ and $g_{\text{true}}$ and $S$ and $S_{\text{true}}$ for a fair comparison. For this purpose, they are scaled so that the sum of all pixels is 1. After normalization, the Sum of squared errors (SSE) is computed. For $k$ and $k_{\text{true}}$, we compute the mean squared error MSE without normalizing the images. For $S$ and $S_{\text{true}}$ the mean value of the SAM is also computed. We add this metric because it is usually used to compare spectra. The spectral angle is more suitable to express changes in chromaticity while the SSE is more suitable to express the mean aspect of the image. SAM is already a scaling invariant metric thus, there is no need to normalize $S$ for this metric.

Table 1 presents the results over the 20 images of the MIT dataset by giving the mean value for all metrics. The quantitative analysis is unfortunately not possible on the CAVE dataset as there is no ground truth.

First of all, the execution time can be analyzed. The time presented in the table is the total time needed to process the whole MIT dataset while parallelizing the processing for the 20 images on a 64 bits Intel Xeon CPU processor at 1.9 GHz, with 12 cores. *Yang*'s method is the fastest, followed by *LS* and *KL*. But it should be noticed that the comparison is not completely fair in the sense that *LS* and *KL*' implementation are Matlab codes, but that call C programs that make them much faster. The proposed method has a good improvement compared to *Krebs*. The proposed method is 32 times faster than the previous one and uses pure programming. The proposed method outperforms the other methods for the four metrics, which was not the case in *Krebs, Benezeth & Marzani (2017)*. Concerning $g$, the proposed method has the lowest error with an SSE of $0.183 \times 10^{-5}$, our previous algorithm

is second with $0.384 \times 10^{-5}$ (almost a factor 2 is gained) and $KL$ is third with $0.504 \times 10^{-5}$. This is consistent with the results shown in the qualitative results section (Fig. 1).

Results on the MSE of $k$ are also favorable to the proposed method with an MSE of $0.007 \times 10^{8}$. It is seven times lower than the previous version and 10 times lower than $LS$ ($0.102 \times 10^{8}$). *Barron*'s method does not return the specular component. The whole image is supposed to be diffuse.

Results for $S$ are also good with a mean angular error of 0.020 radians (1.1 degree) and an SSE of $0.144 \times 10^{-5}$. These results are directly correlated to a better estimation of $g$ and $k$ as $S$ is recovered to respect the equality (23). The quantitative results are consistent with the qualitative analysis. The proposed method outperforms the state of the art by recovering a good estimation of the three components $g$, $S$, and $k$. Moreover, the updated method has a significant gain compared to the one proposed in 2017.

## CONCLUSION

In this paper, a novel method to recover the parameters of the dichromatic model using a single image has been introduced. The algorithm is learning free because it is simply expressed as two independent quadratic programming problems. The method is an updated version compared to the one proposed in 2017. The method is applied to multispectral images and offers a significant gain on RGB images. Two datasets were used for this study, a set of RGB images from the MIT and a set of multispectral images named CAVE. We have assessed our results qualitatively and quantitatively to ensure the quality of the algorithm. The proposed method has better accuracy than recent advances in the field. The good results are coming from the choice of the objective functions, expressed in the logarithmic domain and based on soft and hard constraints. A smoothness constraint helps to improve the quality of the photometric invariants' recovery. The specific architecture of our algorithm, i.e., two simple constrained quadratic programming steps, open opportunities in the field to create memory-efficient and time-efficient algorithms for the recovery of intrinsic images.

### Funding
This study was supported by the French Research National Agency (ANR) program EMMIE under the grant agreement 15-CE17-0015. The funders had no role in study design, data collection and analysis, decision to publish, or preparation of the manuscript.

### Grant Disclosures
The following grant information was disclosed by the authors:
French Research National Agency (ANR) program EMMIE: 15-CE17-0015.

### Competing Interests
The authors declare that they have no competing interests.

## Author Contributions

- Alexandre Krebs conceived and designed the experiments, performed the experiments, analyzed the data, performed the computation work, prepared figures and/or tables, and approved the final draft.
- Yannick Benezeth conceived and designed the experiments, authored or reviewed drafts of the paper, and approved the final draft.
- Franck Marzani conceived and designed the experiments, authored or reviewed drafts of the paper, and approved the final draft.

## Data Availability

The source code to test the method is available in the Supplemental Files.

## Supplemental Information

Supplemental information for this article can be found online at http://dx.doi.org/10.7717/peerj-cs.256#supplemental-information.

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
