# Peer review of "Intrinsic RGB and multispectral images recovery by independent quadratic programming"

_PeerJ Computer Science, doi:10.7717/peerj-cs.256_

## Round 0.1 · original submission · Major Revisions

Please address the issues raised by the reviewers.

Reviewer 1 ·

Basic reporting

The paper is generally well written and clear, however the emphasis on the novelty using compared to the authors' previous work is missing in some parts including the abstract.

Experimental design

The importance of the work can be better shown if the authors explain in more details the advantages of having the objective function in the logarithmic domain which seems to be the main contribution of this work.

Validity of the findings

The results presented are promising, however they lack more discussions. For example, the Specularity can be seen on the apple and pepper reflectance images of the authors' proposed method and previous method, and this point is not discussed.

Moreover, the computation time of the proposed method compared to state of the art approaches is not specified.

Additional comments

The paper is clear with interesting results. However, I suggest to give more emphasis on the originality of
the paper compared to your previous work regarding the logarithmic objective function. Moreover, more details and discussions on the results especially on the errors in estimation is essential in my point of view for a better understanding and impact of this work.

Reviewer 2 ·

Basic reporting

A few typos are still present. I notice numerous errors in section 4.1. “Fig. 1 presents shading images g resulting from all methods on five examples”: where the fixe examples are in figure 1 ? “The next rows are respectively the results given by LS, KL, Barron, Yang…” it’s certainly columns ?

Experimental design

The authors cite a previous work. To my opinion, more information of this previous method must be given to understand this part.

Validity of the findings

It would be advisable to add a table with the computing time.

Additional comments

The paper deals with reflectance, shading and specularity estimation form multispectral image. The process has been evaluated in MIT and CAVE datasets. The authors provide an interesting method and seems to be very efficient. Unfortunately, a few typos are still present.
Major issues (not in priority order):
1°) “3 Method”: this part starts with a citation of a previous work on this topic. To my opinion, more information of this previous method must be given to understand this part.
2°) “4.1 Qualitative results”: I notice numerous errors in this section. “Fig. 1 presents shading images g resulting from all methods on five examples”: where the fixe examples are in figure 1 ? “The next rows are respectively the results given by LS, KL, Barron, Yang…” it’s certainly columns ?
3°) “4.2 Quantitative results”: it would be advisable to add a table with the computing time.

Minor issues:
1°) Equation 1 is the Lambertian BDRF and the reference of 2012 is not appropriate.
2°) Page 2, line 28: “… a special case of the Bidirectional Reflectance Distribution Function (BRDF)…” the BRDF do not include the cos(theta).
3°) page 8, line 216: why “now” ? Only one method is proposed in the paper ?

---

## Round 0.2 · accepted · Accept

You have addressed the issues raised by the reviewers.

Reviewer 2 ·

Basic reporting

Most of the issues raised in the first review phase have been address. The authors have improved the manuscript to a level that could be useful for readers of PeerJ Computer Science.

Experimental design

no comment

Validity of the findings

no comment